# Early Life Adversity, Microbiome, and Inflammatory Responses

**DOI:** 10.3390/biom14070802

**Published:** 2024-07-06

**Authors:** Eléonore Beurel, Charles B. Nemeroff

**Affiliations:** 1Department of Psychiatry and Behavioral Sciences, Miller School of Medicine, University of Miami, Miami, FL 33136, USA; ebeurel@miami.edu; 2Department of Biochemistry and Molecular Biology, Miller School of Medicine, University of Miami, Miami, FL 33136, USA; 3Department of Psychiatry and Behavioral Sciences, Mulva Clinic for Neurosciences, University of Texas (UT) Dell Medical School, Austin, TX 78712, USA; 4Mulva Clinic for Neurosciences, UT Austin Dell Medical School, Austin, TX 78712, USA

**Keywords:** Inflammation, microbiome, childhood neglect

## Abstract

Early life adversity has a profound impact on physical and mental health. Because the central nervous and immune systems are not fully mature at birth and continue to mature during the postnatal period, a bidirectional interaction between the central nervous system and the immune system has been hypothesized, with traumatic stressors during childhood being pivotal in priming individuals for later adult psychopathology. Similarly, the microbiome, which regulates both neurodevelopment and immune function, also matures during childhood, rendering this interaction between the brain and the immune system even more complex. In this review, we provide evidence for the role of the immune response and the microbiome in the deleterious effects of early life adversity, both in humans and rodent models.

## 1. Early Life Adversity

Early life adversity affects 1 out of 10 children (under 18) to the point that these children will experience long-term physical and mental health consequences [1,2], including an increased risk for psychiatric disorders [3], and many other medical disorders including asthma, diabetes, stroke and cardiovascular disease [4]; and an overall marked increased risk of mortality by age 50 [5,6]. The impact of early life adversity likely results from multifactorial mechanisms. The prevailing theory of how early life adversity increases risk for physical and mental illnesses revolves around the body’s response to stress and the idea that there are possible sensitive periods when early life adversity impacts neurodevelopment the most. Physiological changes after acute stress include the release of cortisol and norepinephrine [7], which prepare the body to maximize the response to the threat while minimizing harm and mobilizing the immune system. Early life adversity includes childhood physical or sexual abuse, neglect, and other negative life events (e.g., poverty, loss of a parent, parental psychopathology, bullying, natural disaster, etc.) [8,9]. The high frequency or magnitude of exposure to the stressors during early life adversity dictates the degree and chronicity of subsequent inflammation [10]. Persistent inflammation has a multitude of detrimental effects on the body over time [11,12,13,14,15]. In summary, although the pathophysiological mechanisms responsible for the many detrimental effects of early life adversity on health remain poorly understood, many studies have posited the immune system to be a major mediator. Thus, the release of norepinephrine under threat conditions, mobilizes the immune system to produce cytokines in the damaged tissues to help clear the pathogens and promote wound healing [16]. Simultaneously, the adrenal glands increase the production of cortisol and epinephrine, which initially reinforce the proinflammatory actions of norepinephrine, but after the threat wanes, in contrast, they counteract the effects of norepinephrine [17,18]. From an evolutionary point of view, stress is thought to prepare the body to clear pathogens and enhance tissue healing. Although stress and inflammation are considered culprits during early life adversity, their exact role in critical periods of neurodevelopment remains to be fully elucidated and will not be discussed in this review.

## 2. Immune System

The main function of the immune system is to protect the organism against pathogens (e.g., microbes, toxins, etc.). There are two major immune arms: the innate immune system, which is already active at birth, but weaker than later in life, and the adaptive immune system, which develops as the child is exposed to antigens.

### 2.1. Innate Immune System

The innate immune system comprises neutrophils, monocytes/macrophages, dendritic cells, which react to pathogen invasion by phagocytosing them. Children secrete lower levels of cytokines responsible for the priming of T helper (Th) 1 and CD8 T cell responses, known to eliminate pathogens, compared to adults, whereas their production of cytokines required for Th17 responses, which are required for some bacterial and fungal infection clearance is similar to adults’ function (for review [19]). This suggests an increased susceptibility of children to certain pathogens, which will be discussed later in the review. In summary, the innate immune system is muted at birth to allow all the remodeling during development and to tolerate maternal antigens, and this puts the child at higher risks for bacterial and viral infections and vulnerability to stressors associated with early life adversity.

### 2.2. Adaptive Immune System

If the innate immune system is not able to clear the pathogen, it recruits by producing cytokines and by presenting antigens to the adaptive immune system, whose function is to help the innate immune system clear the pathogen. The adaptive immune system comprises T and B cells. With the help of T cells, B cells produce antibodies to clear pathogens. The first encounter with bacteria is during the passage through the birth canal and then the contact of the skin and lungs with the exterior world. During the child’s life, the adaptive immune system changes and forms immunological memory to protect the child against a new encounter of the same pathogen [20], emphasizing the importance of immunizations [21] to help train the adaptive immune system of children. It is remarkable that a mother can transfer to the fetus immunoglobulins from infections she encountered 20–30 years ago, helping the child cope with infections. There are also many asymptomatic viral infections that induce a strong CD4 and CD8 T cell immune responses, such as cytomegalovirus (CMV) or Epstein–Barr virus (EBV), and any immunosuppressive therapy can reactivate viruses and have dramatic negative consequences for the child. Furthermore, naïve B cells in response to a specific antigen undergo somatic hypermutation, class switching and proliferation, creating B cells producing higher affinity-binding immunoglobulins, a process that takes months and that leads to a better immunity over time [22]. Furthermore, maturation of the T and B cells is critical for immunological memory. As an individual gets older, the repertoire of immune cells experiences maturation. This adds to the already high genetic variability between individuals and explains the considerable variation in the immune response between individuals. For example, Th17, regulatory T cells and NK cells are altered in patients with high liability for psychosis and childhood trauma. Thus, Th17 cell number increases in the high liability group with childhood trauma, regulatory T cells increases, and NK cells decreases in the high liability group and predicted stress experience [23]. Th17 cells previously show a trend to increase with early life adversity [24]. This is consistent with the findings that Th17 cell differentiation is already operational in children, similar to adults [19], that Th17 cells are associated with asthma [25,26], which is linked to early life adversity [27,28] and that Th17 cells promote susceptibility to depression after stress and affect neuronal circuits in autism [29].

Altogether, these findings suggest that any perturbations of the immune system maturation can have consequences for the child health and may explain the increased susceptibility of certain children to mental health and physical health disorders.

## 3. Effects of Early Life Adversity on the Immune System in Humans

Childhood adversity primes immune cells (e.g., monocytes, macrophages) that initiate and sustain inflammation. Thus, maltreated or disadvantaged children are often more exposed to air pollution, high-fat, high-sugar diets or stressors, such as family instability, poverty, and violence [30,31], and these exposures prime monocytes to produce higher levels of cytokines when restimulated ex vivo with microbial products [32,33] and render monocytes less sensitive to inhibition by glucocorticoids [34]. Insensitivity to glucocorticoids has been postulated to be adaptive during acute threats, while in contrast, if sustained, insensitivity to glucocorticoids would promote chronic inflammation involved in emotional and physical health issues [15,35].

### 3.1. Inflammation

#### 3.1.1. Early Life Stress Is Associated with Inflammation

Longitudinal studies in humans have revealed a correlation between early life stress, exposure to trauma and the presence of inflammatory markers in adulthood, such as the circulating acute phase protein, C-reactive protein (CRP), proinflammatory cytokines (interleukin-6, IL-6; IL-1β; tumor necrosis factor, TNF) [8,36,37,38], and the alteration of the leukocyte count [39]. For example, in the Dunedin Multidisciplinary Health and Development Study [40], during which subjects have been followed from birth to adult life, it was found that cumulative exposure to early life maltreatment was associated with greater inflammatory marker levels later in adulthood when controlling for all possible confounding factors [41]. Other studies have confirmed these findings [8,9]. Similar findings were found whether the maltreatment was perpetrated by adults or peers (e.g., bullying). It has been estimated that ~10% of low-grade inflammation (defined as an elevation of 2–3 fold of interleukin-6, tumor necrosis factor and C-reactive protein [8]) in adults may be due to childhood maltreatment [41,42]. Furthermore, early life stress has profound health consequences, because childhood maltreatment is also associated with impaired acquired immunity [43], an enhanced inflammatory response to stressors [11,14,44,45,46] and an increased risk of physical and mental health disorders [47].

The inflammation seems to be triggered by exposure to early life stress, as the increased inflammation was detectable in maltreated children [38], as evidenced by elevated CRP levels in maltreated children. In addition, increased inflammation is associated with the onset of depression [48], reinforcing the importance of inflammation in priming depressive symptoms. Consistent with this, increased inflammation was found in adolescents exposed to adverse childhood experiences [49] and in children with a Child Protective Service-documented history of maltreatment [50].

#### 3.1.2. Comorbidities Render the Association between Early Life Stress and Inflammation More Complex

Comorbid factors such as obesity also render the association between childhood maltreatment and inflammation complex because obesity increases inflammatory markers [51], and an increased risk of obesity is associated with childhood maltreatment [13], raising questions about whether inflammation or obesity emerges first [8,9]. Additional life stresses in adulthood likely also contribute to the alteration of immune measures.

#### 3.1.3. Stress Affects Inflammation

Increased stress in childhood is also associated with increased hypothalamic–pituitary–adrenal (HPA) axis activity, leading to increased production of cortisol [52,53], and cortisol exerts acute anti-inflammatory property [54,55]. Therefore, the concentration of cortisol might dictate the outcome of the inflammatory response. Consistent with this notion, children with supportive caregivers exhibited attenuated HPA axis activity [56]. In adolescents, the HPA axis seems altered [57,58], and the impact of cortisol on the immune response seems to be lessened [59]. This might explain why cytokines are detected more systematically in older children. There is also evidence of profound hyperactivity of corticotropin-releasing factor (CRF)-containing circuits in the brain after early life adversity, and this system is also known to regulate immune function.

#### 3.1.4. Other Factors Also Contribute to Early Life Stress Outcomes on Inflammation

Other behaviors in adolescence might also explain the change in cytokine production, such as changes in diet, absence of exercise, substance and alcohol abuse, and stress associated with risky behaviors [60,61,62,63]. Overall, there is substantial variability in the measured cytokines depending on the type of stress, the age, and the method of assessment of inflammation. No clear picture has emerged about the functionality of the immune system in early life adversity, but growing evidence suggests an integrated, bidirectional interaction between the cortico-amygdala circuit and inflammation to detect threats to well-being and to promote coping through the propagation of inflammation by the monocytes/macrophages [16,64,65]. In addition, the cortico-basal ganglia circuit and reward-related behaviors are also affected by early life adversity. Thus, maltreatment correlated with later reward-processing deficits on behavioral tasks [66,67], and inflammation blunted reward sensitivity, referred to as sickness behaviors, which belong to a set of adaptations to infection mediated by cytokines [68,69]. Consistent with this, inflammation decreased animals’ sensitivity to reward [70]. In addition, prefrontal executive control deficits contribute to neuroimmune responses. The cerebral prefrontal cortex is particularly vulnerable to early life adversity, and reductions in the gray matter volume, cortical thickness, and white matter tract integrity of the cerebral prefrontal cortex were reported in individuals with early life adversity [60,71,72,73,74,75]. Inflammation also interacts with the prefrontal cortex, with reports of smaller prefrontal cortex volume and white matter integrity associated with increased inflammatory markers [76,77]. In summary, the presence of low-grade chronic inflammation in early life adversity seems to expose children to a plethora of health issues across the lifespan [15,78,79,80]. Focusing on mental health issues, it has been proposed that defects in the cortico-amygdala circuit and inflammation manifest in childhood, while reward and executive functions are affected later in life during adolescence and adulthood [81].

### 3.2. Immune Cell Defect Is Associated with Early Life Stress

Both innate and adaptive immune cell functions are affected by early life stress.

#### 3.2.1. Innate Immunity Cell Defects

At the cellular level, a 20% reduction in granulocyte function (assessed by killing ex vivo *Staphylococcus aureus*) has been reported in children of separated or divorced parents [82]. Studies have reported low natural killer (NK) cell activity in adolescents with childhood trauma [83] and in women with breast cancer who had experienced early life adversity [84], increased NK activity in children whose parents reported chronic stress [85], and no change in NK activity in sexually abused adolescents [86]. Therefore, although innate immunity through cytokine production by monocytes seems enhanced, granulocyte and NK functions seem to decrease after early life adversity.

#### 3.2.2. Adaptive Immunity Cell Defects

In term of adaptive immunity, IgE and IL-4 productions were increased in predisposed children to asthma, with parents experiencing chronic stress [27,87,88] suggesting hyperreactive peripheral blood mononuclear cells (PBMCs). It has been reported that early life adversity has a profound impact on the thymus, leading to involution or atrophy of the thymus or the spleen, including spleen atrophy [89], providing a potential basis for a T cell defect because T cells are maturing in the thymus and are housed in the spleen and lymph nodes. Furthermore, research on adolescents and young adults demonstrated that early life adversity events during infancy were associated with changes in specific T cell populations, including T cell immunosenescence [37,90,91], and there was an increase in CD8 T cells relative to CD4 T cells [24,91,92,93]. CMV seropositivity was also associated with the long-term shift in T cells [91,92], showing signs of senescence. All together, these findings point to a reduction in the function of T cells in adolescents with childhood trauma, confirming the significant impact of childhood trauma on immune function.

### 3.3. Early Life Adversity and Increased Susceptibility to Infection

It has been reported that children of parents with psychiatric symptoms are more prone to illnesses [85,94]. Similarly, children who perceive their family as dysfunctional are more susceptible to influenza B infection when compared to children who do not have this perception [95]. Children more sensitive to stress also exhibit higher rates of respiratory infections [96,97], and high parental stress predicts infant wheezing even after controlling for allergen exposure, parental smoking, and susceptibility to respiratory infections [98]. These findings suggest a potential weakening of the immune response in children with high levels of stress.

Lower socioeconomic status has also been associated with an increased risk of infectious disease in adulthood [99,100,101], but this association was absent when the relationship with one parent was supportive [102], reinforcing the importance of a supportive child–parent relationship.

The response to vaccination seems to also be attenuated in children with early life adversity [103,104], consistent with a weakening of the body’s response to vaccines by stress [105]. Although the consequences of a weakened immune response are not clear, it likely contributes to the reactivation of dormant viruses such as the Herpes simplex virus [43,106], Epstein–Barr virus [107,108] and cytomegalovirus [37,94] leading to higher virus-specific antibody titers in children or adolescents with early life adversity. Similarly, sensitization to allergens has been associated with early life adversity [109,110].

In summary, the association between inflammation and early life adversity is complex. Even though there is an increase in proinflammatory cytokines in individuals with early life adversity, it appears that the immune response is less effective than in individuals without early life adversity. This might explain findings of an increased risk of developing depression, cancer, cardiovascular diseases, or other serious illnesses in adulthood of individuals with early life adversity.

### 3.4. Early Life Adversity Affects the Microbiome

The gut microbiome represents a unique community of trillions of microorganisms cohabiting in the gut, including bacteria, fungi and viruses implicated in both mental and physical health. Each individual microbiome is unique and is influenced by the diet, behaviors, and environment. The gut microbiome is critical during the first years of life and may have long-term consequences for health due to its role in shaping the immune system [111]. Prenatal stress impacts the microbiome [112,113,114,115]. Thus, infants with high stress levels [115] or newborns from stressed pregnant mothers [114] exhibited an increased abundance of *Proteobacteria* spp. and a decreased abundance of *Bifidobacterium* spp. Infants with high level of cortisol 3 days postpartum also exhibited increased levels of the Proteobacteria genus *Enterobacteriaceae* [114], though other studies failed to reproduce these findings [116,117]. Changes in the microbiome of infants who experienced maternal prenatal stress have been suggested to shape changes in neurodevelopment [118], which could predispose individuals to psychopathologies.

Changes in the gut microbiome have also been associated with early life adversity and changes in blood T cell subsets in youth [119]. Beta diversity (e.g., differences between the different bacterial communities) and bacterial counts were reported to be reduced in youth exposed to early life adversity [120]. Bacterial changes were also associated with other early life adversity outcomes, such as the activation of the medial prefrontal cortex in emotional faces. This suggests that early life adversity, caused by remodeling the gut microbiome, affects both the immune system and emotional reactivity. *Prevotella*, which is associated with diet [121,122,123] or HIV infection [124], *Bacteroides* and *Coprococcus* levels are augmented in institutionalized youth [119,120]. Similarly, the oral microbiome was affected in individuals with early life adversity, and changes in taxa (e.g., hierarchical subdivisions of the bacteria species) abundance could be detected 24 years after the early life adversity event. Changes in the taxa abundance were associated with early institutionalization and CMV seropositivity [125]. There was also a link between T cell senescence, NK activation status, and changes in the oral microbiome. Although in humans it is difficult to attribute the changes of microbiome and immune function to early life adversity, it points to an interesting association between institutionalization time in the postnatal period, microbiome, and immune changes in youth.

### 3.5. Effects of Early Life Adversity on Pregnancy Outcomes

Early life adversity is associated with a 2-fold increased risk of preterm delivery compared to mothers without early life adversity [126]. This elevated risk for preterm labor is in part attributed to risky behaviors such as smoking, alcohol or substance abuse and mental health conditions such as anxiety and depression [6,126,127,128,129]. In addition, the gestation age, and the weight at delivery of the offspring of mothers with early life adversity are lower [130]. These seem associated with increased maternal cortisol [131] and increased basal levels of IL-6 [132] and are consistent with an adverse effect of inflammation on pregnancy and the developing fetus. Furthermore, inflammation, which is associated with adverse childhood experiences, increases the risk of developing depression during pregnancy [133] and other pregnancy complications, such as gestational diabetes and hypertensive disorders [134].

Parenting may also be difficult for individuals with early life adversity [135], although not all parents with early life adversity mistreat their children. Several studies showed a link between allostatic load (e.g., exposure to stress for a long period of time) and poor parenting [136,137], which is consistent with the negative impact of adverse childhood experiences on cognition and social development [136,137].

Altogether, these findings point to a potential intergenerational transmission of stress and adverse health outcomes. Although intergenerational transmission is difficult to establish methodologically, some researchers have reported evidence for intergenerational transmission of neglect and sexual abuse but not physical abuse [138]. It has been proposed that central and peripheral systems, including neural circuits, the HPA axis, and the microbiome, are possible pathways mediating the effects of early life adversity across generations [139,140,141]. The gut microbiome of pregnant mothers shapes the development of the fetal brain and immune system [142,143]. Recently, a large longitudinal study showed that maternal prenatal anxiety or maternal childhood maltreatment and second-generation exposure to stressful life events are sufficient to impact the gut microbiome composition of the second-generation children at 2 years of age, as well as impacting behaviors (e.g., socioemotional functioning) at 2 and 4 years of age [141]. It also seems that the timing of exposure to early life adversity (prenatal vs. postnatal) has a different impact on the gut microbiome composition, but both are associated with increased inflammation-associated taxa [141]. These findings confirm the notion that early life adversity, by affecting the microbiome could consequently affect the immune system, which in turn could interfere with neuronal circuits, and prime individuals to psychopathologies (Figure 1), although this hypothesis will need to be further tested.

## 4. Modeling Childhood Trauma in Animals and Effects of the Immune System

### 4.1. Proinflammatory Markers

Initial evidence of the impact of childhood experiences on the immune system was reported more than a half century ago in rodents, showing that rats handled before weaning, which promoted bonding with the mother, exhibited better health outcomes [144,145]. Similar behavioral outcomes were reported in non-human primates subjected to early life stress [146,147,148]. Since then, other models have been used to induce early life stress in rodents, such as maternal separation, maternal deprivation, early weaning, or nursery rearing, and those have generated mixed results in terms of inflammatory consequences [149,150,151]. A clearer picture emerges when focusing on maternal separation studies. Indeed, maternal separation induces macrophage activity [152], monocyte-dependent proinflammatory gene upregulation [153], and increases plasma proinflammatory cytokine concentration [154], which is also associated with increased body temperature [155]. There is a lack of evidence for a change in anti-inflammatory cytokines such as IL-10 after maternal separation [156]. Female mice are more prone to increase cytokines after a stressful event than males [157,158]. In the pup brain, maternal separation is associated with a reduction in proinflammatory mediators in the hippocampus, such as lipopolysaccharide-binding protein [159] and of microglia in the midbrain [160]. In contrast, an increased level of the interleukin 1 receptor on synapses [161], an increased number of microglia [162], and an increased activated state [163] are observed in the adult brains of rodents subjected to maternal separation, suggesting possible time-sensitive windows for the effects of inflammation on neuronal development. Furthermore, maternal separation primes rodents for even more robust responses when encountering subsequent stress or immune challenges. Thus, rodents exposed to maternal separation exhibit a greater increase in core temperature after re-exposure to maternal stress [155], greater cytokine production to a viral infection [164], and greater microglia activation after stress [163].

### 4.2. Sex Differences

In addition, an increasing body of evidence suggests that male and female rodents are not equally emotionally affected by early life adversity [165,166,167,168,169]. Early life adversity induces social deficits in male offspring but increases anxiety in female offspring [167,170,171,172,173,174,175,176], which seems to correspond to differential microbial and immune changes between males and females. Associated with these differential outcomes in behaviors, males exhibit predominantly a down regulation of overall gene expression in the medial prefrontal cortex, while overall gene expression is upregulated in females [177], suggesting that the prefrontal cortex executive circuit is not affected similarly in males and females after early life adversity.

### 4.3. Microbiome

Because the stress response is shaped by the microbiome [178,179,180,181] and vice versa, the microbiome is shaped by the stress response [118,182,183,184,185,186], prenatal stress in animals affects the microbiome [187,188] similarly to the findings in humans. In rats, an increased *Prevotella* level is associated with early life adversity [189]. In male mice, *Lachnospiraceae* was reported to be highly sensitive to multiple early life stresses, while in female mice, only *Lactobacillus* and *Mucispirillum* are affected [177]. Maternal separation increases *Bacteroides*, *Lachnospiraceae* and *Clostridium XIVa* spp. in both mice and rats [184,190,191,192]. Associated with a change in the microbiota composition, there is also the presence of gut dysbiosis and associated gut leakiness in animals exposed to early life stress [186,189,191,193,194,195]. Because the gut microbiome is critical during brain development, it has also been proposed that changes in the gut microbiome could affect brain development during early life adversity [196]. Early life stress affects the medial prefrontal cortex, a brain area involved in the regulation of emotional behavior [197,198]. Expression of immediate early genes such as *Arc*, *Egr4*, and *Fosb* in the medial prefrontal cortex is also regulated by the gut microbiota [190,199,200], suggesting a convergence of effects of early life adversity and the microbiota on the medial prefrontal cortex.

Altogether, in animals, early life stress promotes inflammation and alteration of the microbiome in a sex-dependent manner, leading to long-lasting behavioral changes.

## 5. Conclusions

Overall, early life adversity has a negative impact on health, priming individuals for psychopathologies and other medical disorders. It is now unequivocally recognized as one of the preeminent factors in adult morbidity and mortality. The precise cause of the increased vulnerability remains unclear, although the inflammation has been proposed to be one of the major culprits. In addition, patients with mood disorders, including depression and bipolar disorder, who have childhood maltreatment histories have a more pernicious disease course and are relatively treatment resistant, in part due to increases in inflammatory status. Moreover, children with early life adversity are more susceptible to infections and exhibit immune responses that are disrupted throughout the individual’s life. Although the precise causes of the immune dysregulation remain obscure, some have proposed that infection with CMV could induce immunosenescence, rendering the immune system less efficient at clearing insults. These perturbations in the immune system might also be amplified by a less diverse microbiome due to poorer nutrition or hygiene and a lower socioeconomic environment. Gut dysbiosis and leakiness lead to increased neuroinflammation, which has detrimental effects on various neuronal circuits, such as the cortico-amygdala circuit, cortico-basal ganglia circuit and prefrontal executive control circuit, and may lead to behavioral changes associated with the development of psychopathologies. However, evidence is still lacking to show a causal effect of early life adversity on the immune system and microbiome in humans, warranting additional research on the topic.

## Figures and Tables

**Figure 1 biomolecules-14-00802-f001:**
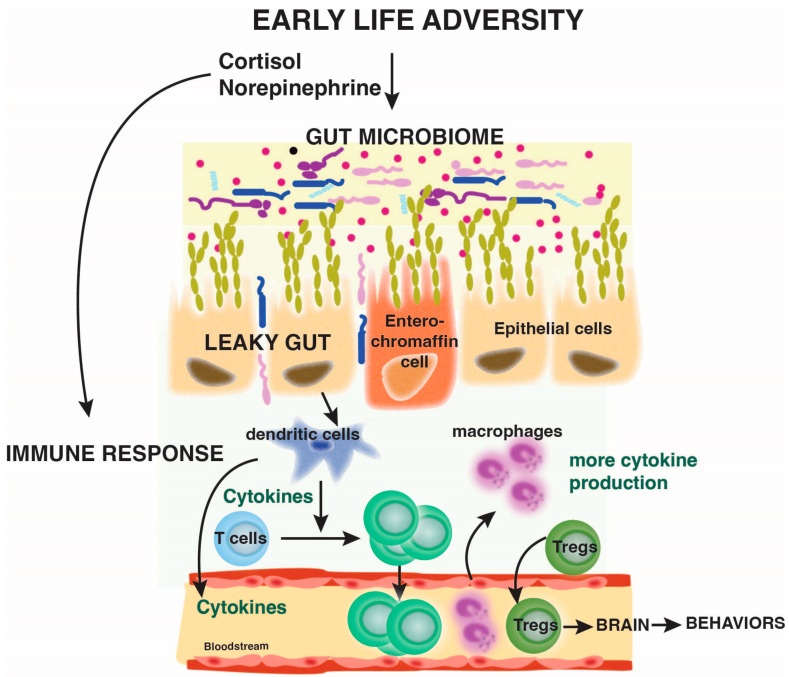
Speculative integrative view of the role of early life adversity on the microbiome and the immune response to mediate behaviors. Early life adversity triggers changes in the microbiome and in the immune system. It is likely that cortisol and norepinephrine mediate some of the effects of early life adversity on the microbiome and the immune system. Alteration of the microbiota composition is also associated with a leaky gut, possibly contributing to the activation of immune cells to produce cytokines in the lamina propria. Increased sustained cytokine production, in conjunction with bacterial encounters due to leaky gut, might be responsible for the differentiation of T helper cells (e.g., Th17 cells or Tregs). Th17 cells are known to promote susceptibility to depressive-like behaviors. Furthermore, the constant presence of antigens might contribute to immune senescence and the increased susceptibility to infections in children with early life adversity. These interactions are speculative, and further research is required to fully elucidate the effects of early life adversity on immunity.

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
