# Peer review of "Early Life Adversity, Microbiome, and Inflammatory Responses"

_biomolecules, 2024, doi:10.3390/biom14070802_

Round 1

Reviewer 1 Report

Comments and Suggestions for Authors

The manuscript addresses the interaction of inflammation, microbiome, and early adversity. The inclusion of the microbiome is unique. Overall, the review presents a significant amount of information regarding the immune system. However, the presentation of the information is disorganized and adversely affected by the inclusion of speculative relationships between relevant factors. The manuscript needs structure before it is suitable for publication.

The initial primer on the development of the immune system (adaptive vs acquired) is rarely presented in our field but offers a nice overview.

The next section of early life adversity on the immune system, however, is choppy.  Perhaps the inclusion of additional subheadings would aid the organization. 

                  Section 3.1 Proinflammatory markers: includes a brief listing of mechanistic changes but then skips around with evidence from different populations to outcomes to comorbidity factors. The subsection on the HPA axis is quite unclear and might fare better at the beginning of this section on mechanism. Again, back to comorbidity factors.  Then to brain changes. This is a lot of territory without a clear conclusion.

The other subsections have similar issues that could be helped with topic sentences and more subheadings.

The other general comment has to do with loose associations and speculations that are made. For example, the following passage tries too hard to incorporate any data that might be relevant, and as a result, the point is lost:

                  "Early life adversity also affected the medial prefrontal cortex activation to emotional faces, which was also associated with bacteria levels, suggesting that early life adversity by remodeling the gut microbiome affects both the immune system and emotional reactivity."

A few sentences are rather complex and should be simplified for readability.

1. Section 2.2. Sentence beginning: "Of particular note,..."

2. Section 3.1. Sentence beginning: "Of note, ..."

3. Section 3.1. Sentence beginning: "Of particular note, the increased ..." This sentence builds on the last, but adds the detail of CRP. Better phrased as: "For example, elevated levels of CRP in maltreated children are associated with the onset of depression."

(A trend is emerging- when extra information is inserted that is introduced "of note," the sentence is not well-integrated with the rest of the text.

Section 3.2 Immune cell defect would be better placed after the section on the immune system rather than after the inflammatory markers.

Taxa abundance needs a definition for those less versed in microbiome jargon.

Figure 1 needs a much more descriptive figure caption.

Author Response

We thank the reviewers for their comments which strengthened the manuscript. Our responses are in blue below.

The manuscript addresses the interaction of inflammation, microbiome, and early adversity. The inclusion of the microbiome is unique. Overall, the review presents a significant amount of information regarding the immune system. However, the presentation of the information is disorganized and adversely affected by the inclusion of speculative relationships between relevant factors. The manuscript needs structure before it is suitable for publication.

The initial primer on the development of the immune system (adaptive vs acquired) is rarely presented in our field but offers a nice overview.

The next section of early life adversity on the immune system, however, is choppy.  Perhaps the inclusion of additional subheadings would aid the organization. 

                  Section 3.1 Proinflammatory markers: includes a brief listing of mechanistic changes but then skips around with evidence from different populations to outcomes to comorbidity factors. The subsection on the HPA axis is quite unclear and might fare better at the beginning of this section on mechanism. Again, back to comorbidity factors.  Then to brain changes. This is a lot of territory without a clear conclusion.

RESPONSE: Section 3.1. has been reorganized.

The other subsections have similar issues that could be helped with topic sentences and more subheadings.

 RESPONSE: other subsections have been reorganized.

The other general comment has to do with loose associations and speculations that are made. For example, the following passage tries too hard to incorporate any data that might be relevant, and as a result, the point is lost:

                  "Early life adversity also affected the medial prefrontal cortex activation to emotional faces, which was also associated with bacteria levels, suggesting that early life adversity by remodeling the gut microbiome affects both the immune system and emotional reactivity."

 RESPONSE: This sentence was modified to “Bacterial changes were also associated with other early life adversity outcomes, such as the activation of the medial prefrontal cortex to emotional faces.”

A few sentences are rather complex and should be simplified for readability.

  1. Section 2.2. Sentence beginning: "Of particular note,..."
  2. Section 3.1. Sentence beginning: "Of note, ..."
  3. Section 3.1. Sentence beginning: "Of particular note, the increased ..." This sentence builds on the last, but adds the detail of CRP. Better phrased as: "For example, elevated levels of CRP in maltreated children are associated with the onset of depression."

 RESPONSE: These sentences have been simplified.

(A trend is emerging- when extra information is inserted that is introduced "of note," the sentence is not well-integrated with the rest of the text.

Section 3.2 Immune cell defect would be better placed after the section on the immune system rather than after the inflammatory markers.

 RESPONSE: 3.1. is now labelled inflammation, and 3.2. now logically follows the inflammation section.

Taxa abundance needs a definition for those less versed in microbiome jargon.

RESPONSE: taxa abundance was defined.

Figure 1 needs a much more descriptive figure caption.

RESPONSE: the figure legend was expanded.

Reviewer 2 Report

Comments and Suggestions for Authors

This review article takes on a very interesting and important topic; the role of the inflammation and the gut microbiome as mechanisms of the negative effects of adversity on risk for psychopathology. 

The article cites many of the key papers in this domain and ties them together to propose an interesting and empirically testable mechanistic risk pathway. 

Despite these strengths, there are several areas where the paper could be improved.  The first and perhaps most important is that the  review does not provide a “critical” review in which findings from the extant literature are placed in an order of importance based on the strength of the scientific design.  Without this, it is very difficult to determine the relative power of the various associations cited and this obscures where the literature is strong and where it is weak.  Adding this level of analysis would be critical to make sense out of the more preliminary and potentially spurious findings and those that are more solid.  This is particularly important in the novel area of the gut microbiome where data complexity is high and greater statistical controls are needed. 

Next and related to the above is the complexity of the adversity area and it’s relation to the factors of interest here.  As adversity is a multi-factorial risk associated with increased toxic exposures, changes in diet and absence of basic health protections, how these confounds impact or are considered in the associations with inflammatory factors and the gut microbiome is absolutely critical.  This issue while mentioned is not adequately taken into account in the presentation of the extant literature.   For example, in associations between adversity and gut microbiome, the role of diet (e.g. breast feedings vs. bottle) is central and may explain some the effects found. 

Another key issue of importance is that the role of development is given very cursory consideration.  The focus seems to be  on “childhood” adversity but infant studies are mentioned and combined with studies from adolescents.  The issue of timing and sensitive periods when adversity may have more powerful impacts is increasingly important and evident in the literature.  Greater specificity with regard to developmental timing of exposures would greatly enrich this review.

Last, the authors tie in the adversity literature with the stress and cortisol and HPA axis literature.  While this is logical and an area of interest, the connection here is far more speculative.  Related to this issue above of adding more nuance to the power of each paper/finding, this is in need for greater contextualization given the lack of specificity in this domain that has plagued it for years. 

Author Response

We thank the reviewers for their comments which strengthened the manuscript. Our responses are in blue below.

This review article takes on a very interesting and important topic; the role of the inflammation and the gut microbiome as mechanisms of the negative effects of adversity on risk for psychopathology. 

The article cites many of the key papers in this domain and ties them together to propose an interesting and empirically testable mechanistic risk pathway. 

Despite these strengths, there are several areas where the paper could be improved.  The first and perhaps most important is that the  review does not provide a “critical” review in which findings from the extant literature are placed in an order of importance based on the strength of the scientific design.  Without this, it is very difficult to determine the relative power of the various associations cited and this obscures where the literature is strong and where it is weak.  Adding this level of analysis would be critical to make sense out of the more preliminary and potentially spurious findings and those that are more solid.  This is particularly important in the novel area of the gut microbiome where data complexity is high and greater statistical controls are needed. 

RESPONSE: We provided a more careful review of the literature.

Next and related to the above is the complexity of the adversity area and it’s relation to the factors of interest here.  As adversity is a multi-factorial risk associated with increased toxic exposures, changes in diet and absence of basic health protections, how these confounds impact or are considered in the associations with inflammatory factors and the gut microbiome is absolutely critical.  This issue while mentioned is not adequately taken into account in the presentation of the extant literature.   For example, in associations between adversity and gut microbiome, the role of diet (e.g. breast feedings vs. bottle) is central and may explain some the effects found. 

RESPONSE: Diet has been taken into consideration. We do not think that the mode of administration of milk affects the microbiome, but we agree that the composition of the diet/milk is critical to shape the microbiome.

Another key issue of importance is that the role of development is given very cursory consideration.  The focus seems to be  on “childhood” adversity but infant studies are mentioned and combined with studies from adolescents.  The issue of timing and sensitive periods when adversity may have more powerful impacts is increasingly important and evident in the literature.  Greater specificity with regard to developmental timing of exposures would greatly enrich this review.

RESPONSE: From the literature, it seems that adversity regardless of the neurodevelopmental timing increases inflammation, and from the point of view of inflammation, there does not seem to have sensitive periods.

Last, the authors tie in the adversity literature with the stress and cortisol and HPA axis literature.  While this is logical and an area of interest, the connection here is far more speculative.  Related to this issue above of adding more nuance to the power of each paper/finding, this is in need for greater contextualization given the lack of specificity in this domain that has plagued it for years. 

RESPONSE: We clarified that some of the connections are speculative, and have added nuances to each literature finding.

Round 2

Reviewer 1 Report

Comments and Suggestions for Authors

The addition of the subtitles helped significantly with readability.

Author Response

thank you for your help strengthening the manuscript.

Reviewer 2 Report

Comments and Suggestions for Authors

So unfortunately I do not find in the revised document meaningful
revisions that address my critique. I see yellow highlighted sections
that address sensitive periods but not other substantive critiques
particularly about the need for a more critical review of the
literature on the microbiome and the need to modify conclusions about the role of the microbiome in brain.

Author Response

We thank the reviewer for his helpful comments.